# Novel Approach for the Synthesis of Chlorophosphazene Cycles with a Defined Size via Controlled Cyclization of Linear Oligodichlorophosphazenes [Cl(PCl_2_=N)_n_–PCl_3_]^+^[PCl_6_]^−^

**DOI:** 10.3390/ijms22115958

**Published:** 2021-05-31

**Authors:** Mikhail Gorlov, Nikolay Bredov, Andrey Esin, Igor Sirotin, Mikhail Soldatov, Volodymyr Oberemok, Vyacheslav V. Kireev

**Affiliations:** 1Faculty of Petrochemistry and Polymer Materials, Mendeleev University of Chemical Technology of Russia, Miusskaya sq. 9, 125047 Moscow, Russia; mikgorlov@gmail.com (M.G.); nbredov@muctr.ru (N.B.); aesin@muctr.ru (A.E.); isirotin@muctr.ru (I.S.); kireev@muctr.ru (V.V.K.); 2Taurida Academy, Department of Biochemistry, V. I. Vernadsky Crimean Federal University, Prospekt Akademika Vernadskogo 4, 295007 Simferopol, Russia; genepcr@mail.ru

**Keywords:** phosphazenes, cyclization, controlled cycle size, living cationic polymerization, hexamethyldisilazane

## Abstract

Despite a significant number of investigations in the field of phosphazene chemistry, the formation mechanism of this class of cyclic compounds is still poorly studied. At the same time, a thorough understanding of this process is necessary, both for the direct production of phosphazene rings of a given size and for the controlled cyclization reaction when it is secondary and undesirable. We synthesized a series of short linear phosphazene oligomers with the general formula Cl[PCl2=N]n–PCl_3_^+^PCl_6_^–^ and studied their tendency to form cyclic structures under the influence of elevated temperatures or in the presence of nitrogen-containing agents, such as hexamethyldisilazane (HMDS) or ammonium chloride. It was established that linear oligophosphazenes are inert when heated in the absence of the mentioned cyclization agents, and the formation of cyclic products occurs only when these agents are involved in the process. The ability to obtain the desired size phosphazene cycle from corresponding linear chains is shown for the first time. Known obstacles, such as side interaction with the PCl_6_^–^ counterion and a tendency of longer chains to undergo crosslinking elongation instead of cyclization are still relevant, and ways to overcome them are being discussed.

## 1. Introduction

Linear high molecular weight phosphazenes are of great interest due to their unique properties such as high flexibility of the polymer chain, low glass transition temperature, high thermal stability and high values of quadratic electro-optic effect [1,2]. Thereby, these polymers find applications as elastomers [3], solid polyelectrolytes [4], photosensitive materials [5], gas separation membranes [6], liquid crystals [7,8], non-linear optic materials [9], sensors and actuators [10], extractants [11] etc.

Phosphazene structures and their properties can be easily tuned due to the ease of backbone derivatization by the nucleophilic substitution of chlorine atoms with various organic groups [4,12] (Figure 1).

Since the synthesis of high polyphosphazenes is quite complicated, it is a common approach to use cyclic compounds, mainly hexachlorocyclotriphosphazene (HCP), as the relevant models [13] suitable for the extrapolation of the functional properties and reaction activity of linear polymers. Of course, cyclophosphazenes are in demand as self-sufficient materials as well: for example, as curing agents for polymer resins [14,15,16]; as modifiers for polymers to improve their inflammability [17,18] and mechanical characteristics [17,19]; as building blocks for porous materials [20,21]; as cores of star-shaped polymers [22,23,24,25] and dendrimers [26,27,28] and much more. The size of a cycle plays an important role when phosphazenes are used as coordination ligands for metal ions [2,29,30] or as building blocks in the preparation of porous materials [31].

The most comprehensively studied HCP was first obtained in 1843 by von Liebig, as a side compound formed in very low yield from the interaction of phosphorus pentachloride with ammonia [32]. In 1896, Stokes [33] isolated and identified HCP, which he obtained as the target product in 27% yield by reaction of NH_4_Cl with PCl_5_ in a sealed vessel at 150–200 °C. Since the work of Schenk and Romer [34], the interaction of phosphorus pentachloride and ammonium chloride has usually been carried out in a solvent, usually sim-tetrachloroethane and, later, in less toxic and cheaper chlorobenzene. The rate of the ammonolysis reaction is highly temperature dependent. For the thorough completion of the interaction of PCl_5_ and NH_4_Cl in boiling chlorobenzene (b.p. 131 ° C), 25–30 h are required, or, when using *sym*-tetrachloroethane (b.p. 147 °C), only 7–8 hours [35]. Carrying out the reaction in anhydrous pyridine, which acts simultaneously as a solvent and an acceptor of the hydrogen chloride formed during the reaction, reduces the reaction time to 1 h with a yield of the target HCP of 80%. As ammonium chloride is practically insoluble in organic solvents, its solution interacts with PCl_5_ at the phase boundary. Therefore, when using finely dispersed NH_4_Cl, both the reaction rate and the yield of cyclic chlorophosphazenes increase due to a local increase in the concentration of ammonium chloride [35].

Becke-Goehring et al. were the first to propose a probable mechanism for the formation and growth of the phosphazene chain [36]. According to their theory, phosphorus pentachloride is present in the system in the form of an ion pair, while ammonium chloride dissociates into NH_3_ and HCl (Figure 2a,b). In the first stage of the reaction, a nucleophilic attack of a positively charged phosphorus atom with an ammonia molecule occurs (Figure 2c). The result is a highly reactive trichlorophosphoraneimine Cl_3_P=NH, which interacts with the next ion pair [PCl_4_]^+^[PCl_6_]^−^ (Figure 2d). Trichlorophosphazotrichlorophosphonium hexachlorophosphate [Cl_3_P=N-PCl_3_]^+^[PCl_6_]^−^ is an intermediate product formed at the initial stage of the process, but it is stable in anhydrous medium and was isolated in pure form [35]. [Cl_3_P=N-PCl_3_]^+^[PCl_6_]^−^ is poorly soluble in non-polar solvents, as a result of which the growth process of the phosphazene chain becomes heterophase and slows down after the Figure 2d stage. As the chain grows according to a schema similar to Figure 2d, and the length of the cation increases, its solubility increases, and further interaction with ammonia occurs in solution. Cyclic products are formed during the cyclization of linear oligophosphazenes under the action of ammonium chloride (Figure 2e).

Emsley et al. [37] presented an alternative schema for the cyclization of growing phosphazene chains (Figure 2f), which does not require the participation of NH_4_Cl and is accompanied by the elimination of the [PCl_4_]^+^ cation. The Emsley schema is in poor agreement with the results of Paddock’s studies, which indicate a strict dependence of the yield of cyclic reaction products on the presence of an excess of ammonium chloride in the system [38]. Thus, it has been shown that the yield of a mixture of cyclophosphazenes can be increased from 67%, typical for the classical reaction of partial ammonolysis of PCl_5_, to 93% by maintaining an excess of NH_4_Cl in the reaction medium due to the gradual introduction of phosphorus pentachloride. Emsley [37] also found that the formation of the hexamer (N=PCl_2_)_6_ occurs before the appearance of a pentamer in the system, and suggested that the formation of higher cyclic phosphazenes at high temperatures occurs not through intramolecular cyclization but as a result of the intermolecular interaction of two smaller cycles: 2 (N=PCl_2_)_3_ → (N=PCl_2_)_6_.

In 2014, Bowers et al. [39] separated the products of the ammonolysis reaction by liquid chromatography, which made it possible to isolate the cycles (NPCl_2_)*_n =_*
_5–9_ and identify them separately from each other using ^31^P NMR spectroscopy and mass spectrometry. The obtained values of chemical shifts on ^31^P nuclei, given in Table 1, contradicted the generally accepted data, in particular, the fact that the hexamer signal turns out to be shifted to the region of a weaker field relative to the pentamer singlet. Taking these data into account, the results obtained by Emsley indicate a gradual increase in the size of the formed cycles, with an increase in the duration of the reaction. Thus, cyclophosphazenes are most likely formed by intramolecular cyclization.

A significant step in the development of the chemistry of phosphazenes was the discovery of the method of living cationic polymerization of phosphoraneimine in the presence of Lewis acids, which made it possible to controllably obtain linear compounds of a given length with a narrow molecular weight distribution at ambient temperature [40]. Later, Allcock showed [41] that HCP may also be obtained by the interaction of PCl_5_ with tris-(trimethylsilyl)amine N (SiMe_3_)_3_ (Figure 3a), with a yield of up to 70%, which is comparable to the yield of the classical reaction, but the process proceeds under milder conditions.

De Jaeger et al. [42] investigated the possibility of obtaining cyclic and linear phosphazenes by the interaction of salts [Cl_3_P=N–PCl_3_]^+^[PCl_6_]^−^ and [Cl_3_P=N–PCl_3_]^+^Cl^−^ with hexamethyldisilazane (HMDS). The authors assumed that the process of phosphazene chain growth begins with the replacement of the terminal –PCl_3_^+^ cation of the growing chain with the =N–SiMe_3_ group. Subsequent condensation of the formed linear oligomers can occur both intermolecularly and intramolecularly with the formation of linear polyphosphazenes and cyclic compounds, respectively. In this way, with an excess of HMDS, a cyclic trimer can be obtained with a yield of up to 50%.

Thus, at present, no methods are known that allow for achieving the predominant content of higher cyclic homologues of a given structure in the product while excluding the formation of (N=PCl_2_)_3_ and (N=PCl_2_)_4_. It is only possible to slightly increase the yield of cyclic homologues of a certain size in the presence of some catalysts for the ammonolysis reaction (mainly anhydrous chlorides of metals such as magnesium, tin (IV), titanium (IV), molybdenum (V), zinc and cobalt [35,43,44,45]).

The above-mentioned facts show that the development of novel methods for the controlled synthesis of linear and cyclic phosphazenes still remains an actual challenge, and is necessary for both fundamental and applied aspects. Although the synthesis of HCP is not a technologically difficult task, and a lot of synthetic routes—traditional from PCl_5_ and NH_4_Cl, and modern ones based on the reaction of PCl_5_ with N(SiMe_3_)_3_ [41]—are known, the final product usually contains impurities of side products and demands a multistep purification.

On the contrary, the higher phosphazene cycles nowadays can be obtained only by poorly controlled methods, which in turn lead to irreproducible results and the impossibility of obtaining quantitative yields.

Despite the large number of works dedicated to phosphazene chemistry, the mechanism of cyclic compounds formation is still not unequivocally approved. Meanwhile, the complete understanding of this process is vital for producing cycles with a defined size on the one hand, and for the prevention of undesired cyclization on the other. Here, we report the synthesis of the short linear phosphazene oligomers Cl[PCl_2_=N]_n_–PCl^3+^PCl^6−^ via living cationic polymerization of Cl_3_P=NSiMe_3_ and the investigation of their ability for intramolecular cyclization under thermal treatment, or by the reaction with hexamethyldisilazane (HMDS) HN(SiMe_3_)_2_ or ammonium chloride NH_4_Cl. A possible schema for the formation of cyclic and linear phosphazenes is proposed and discussed in comparison with the known literature data.

## 2. Experimental Procedure

### 2.1. Materials

Phosphorus pentachloride, hexamethyldisilazane (HMDS) were purchased from Sigma-Aldrich (St. Louis, MO, USA) and used as received. Solvents were purified according to known methods; their physical characteristics corresponded to the literature data [46].

### 2.2. Synthesis

#### 2.2.1. General Procedure of Synthesis of Linear Oligodichlorophosphazenes

One hundred milliliters of dichloromethane and HMDS under argon flow were placed into three-necked flask, equipped with magnetic stirrer and reflux condenser. The solution was stirred for 15 min at −55 °C and then 5 g (0.024 mol) of PCl_5_ were added in one portion. The reaction mixture was then stirred for another 15 min at −55 °C and then the temperature was raised over 2 h until 0 °C, and mixture was stirred at this temperature for another 1 h. After that, the mixture was naturally heated until room temperature and stirred for 2 h. After ending of reaction, the mixture was filtered under argon flow from the precipitated NH_4_C, and all volatile products were removed by vacuum rotary evaporation. The resulted product looked like viscous transparent greenish liquid. Loadings of HMDS and yields of products are given in Table 2.

#### 2.2.2. General Procedure for Cyclization Reaction

The calculated amount of HMDS was added under argon flow to reaction mixture, containing 2 g of linear oligodichlorophosphazene with specified chain length. After stirring at room temperature for 1.5 h, the mixture was filtered, and all volatile compounds were removed by rotary evaporation. Loadings of HMDS and yields of products are given in Table 3.

#### 2.2.3. Synthesis of Phenoxycyclotriphosphazenes

In a three-necked flask with reflux condenser and magnetic stirrer were added, under argon flow, 100 mL of dioxane, 3.89 (0.0414 mol) of phenol and 0.95 (0.0414 mol) of sodium. After full sodium dissolution, a dropwise 30 mL of solution of 2 g of cyclic chlorophosphazene in dioxane were added to the resulted mixture. Then, the mixture was stirred for 48 h at 100 °C. After that, the mixture was cooled to room temperature and poured to water. The precipitated product was dissolved in chloroform and washed subsequently with 3% alkaline solution, 10% NaHCO_3_ solution and water till neutral medium. Then the chloroform solution was dried with Na_2_SO_4_ and the solvent was removed by rotary evaporation. Yield of product after drying was 74–83%, depending on chain length.

### 2.3. Characterization

^31^P NMR spectra were recorded on “Bruker AMX-360” spectrometer (145.7 MHz) with the use of solvents CDCl_3_ and acetone-d_6_, and with the use of 80% H_3_PO_4_ as internal standard. MALDI-TOF spectra were recorded on “Bruker Auto Flex II” spectrometer.

## 3. Linear Chains Preparation

A significant step was reached in phosphazene chemistry with the development of the Lewis acid catalyzed living cationic polymerization of phosphoranimines, usually with N-(trimethylsilyl)trichlorophosphoranimine Cl_3_P=NSiMe_3_, which allowed for the obtaining, at room temperature, of linear compounds with defined lengths and narrow molecular weight distributions [40]. Cl_3_P=NSiMe_3_ preparation is complicated by its instability, its tendency toward spontaneous polymerization and its complex purification procedure. 

Within this work, based on previously developed methods [47], one-pot approach for the preparation of linear phosphazenes by direct reaction of PCl_5_ with HMDS was utilized. This allowed the need for pure phosphoranimine isolation to escape. The presence of unreacted excess PCl_5_ during the synthesis of monomer led to spontaneous growth of phosphazene chain when the reaction mixture was heated up to room temperature. The polymerization degree was controlled by changing the molar ratio between initial reagents PCl_5_ and HMDS. Hence, this one-pot synthesis can be divided into two stages: the formation of monomer Cl_3_P=NSiMe_3_ at –55 °C and its further living cationic polymerization, initiated by the residual unreacted PCl_5_ and activated when temperature is naturally increased up to 20 °C. The analogous method was described in work [48], where the monomer Cl_3_P=NSiMe_3_ was at first obtained from PCl_3_ [49] and then initiator PCl_5_ was added to reaction mixture to start the polymerization process. In the present work, PCl_5_ was added in one portion and its initiation ability was regulated only by temperature of the process.

As the polymerization is initiated by the ionic form of PCl_5_ built by two molecules, with *m* moles of added PCl_5_ as an initiator, the *m/2* of active centers for polymerization will be formed (Figure 4a). In general, the formation of monomer Cl_3_P=NSiMe_3_ can be demonstrated by the schema in Figure 4b. If *m* moles of PCl_5_ or *m*/2 moles of [PCl_4_]^+^[PCl_6_]^−^ are taken, the polymerization degree n of final linear product will be as follows:*n* = 3/(*m*/2) = 6/*m*,(1)
where 3—total molar amount of Cl_3_P=NSiMe_3_ monomer, *m*/2—the number of polymerization active centers (Figure 4c).

The general scheme of polymerization is shown in Figure 4d and represents combination of Figure 4b,c.

Postulating the mechanism of phosphazene chain growth when Cl_3_P=NSiMe_3_ is polymerized by the initiation with PCl_5_ ([PCl^4+^][PCl^6−^]) (Figure 4e) is an important challenge. It probably starts with nitrogen atom of compound 1 coordinating with phosphonium cation 2 and the formation of intermediate structure 3 (Figure 4f). This intermediate structure is stabilized due to the delocalization of positive charge within four atoms of the formed cyclic structure. When this cycle collapses, Me_3_SiCl evolves and a new intermediate structure 4 with positively charged tetracoordinated phosphorus atom is obtained. This compound quickly transforms to more stable form with terminal trichlorophosphonium cation, stabilized with PCl^6−^ (Figure 4g). This compound and its higher homologues react in analogous manner with compound 1 and continue the chain growth (Figure 4e).

As the terminal trichlorophosphonium cations are the active centers of polymerization, the molecular weight of the formed polymer can be regulated by the amount of the initial PCl_5_. The dependence of molecular weight of linear polyphosphazene from amount of initial PCl_5_ used for initiation (*n* = f(*m*)) has a hyperbolic character, and is shown in Figure 5. In order to obtain a number of phosphazene compounds [Cl(Cl_2_P=N)*_n_*PCl_3_]^+^[PCl_6_]^−^ of various chain lengths, we repeatedly conducted the reaction between PCl_5_ and HMDS with various molar ratios (Table 3). The average degree of polymerization was determined by ratio of integral intensities for signals of terminal and backbone phosphorus atoms in ^31^P NMR spectra, and the results were in a good agreement with the theoretical calculations. The one-pot synthesis proposed in the presented work is promising for the preparation of linear phosphazenes, with defined chain length and polymerization degree not higher than 20. It can probably be explained by decreasing of chain growth rate due to the decrease of electrophilicity and activity of the growing macrocation [50].

Nowadays, there is no effective method which allows one to obtain predominantly higher cyclic homologues and exclude the formation of low cycles (N=PCl_2_)_3_ and (N=PCl_2_)_4_. Some catalysts, as usual anhydrous chlorides of such metals as Mg, Sn (IV), Ti (IV), Mo (V), Zn [35,43,44] and Co [45], definitely have some influence, and variation of their amount can slightly increase the yield of cyclic products with defined sizes. That is why it was interesting to evaluate the tendency of compounds [Cl(PCl_2_=N)*_n_*PCl_3_]^+^[PCl_6_]^−^ (where *n* = 2–7) toward spontaneous cyclization under various conditions.

## 4. Thermal Cyclization

Emsley and coworkers [37] proposed a schema for phosphazene chains cyclization which includes the elimination of terminal [PCl]^4+^ cation. However, Allcock showed that [Cl(PCl_2_=N)_4_PCl_3_]^+^[PCl]^6−^ does not form cyclic products under refluxing in dichloromethane for 24 h [41].

In the current work, the linear phosphazene compounds with various chain lengths were refluxed at 131 °C for 4 h in chlorobenzene, and the formation of cyclic product was not observed in any case. Therefore, the schema proposed by Emsley is highly doubtful and cyclization with the elimination of terminal –PCl_3_^+^ group does not occur. Nevertheless, the possibility of the reaction of two cycles with the formation of the bigger cycle still cannot be excluded, especially in light of work by Sirotin [43], where such reactions as 2 (N=PCl_2_)_3_ → (N=PCl_2_)_6_ and 2 (N=PCl_2_)_4_ → (N=PCl_2_)_8_ were observed in mass-spectra after electron beam ionization.

## 5. Cyclization in the Presence of N-Containing Agents

The cyclic products were obtained according to Figure 3a from preliminary synthesized oligodichlorophosphazenes with various chain length by addition of stoichiometric amounts of HMDS (Table 3). NMR data showed the complete cyclization of all phosphazene chains.

According to the following schema, the formation of cycles is accompanied by the enlargement of chain length on one -PCl_2_=N- unit due to the introduction of nitrogen atom from HMDS.

Hence, in the case of [Cl(PCl_2_=N)_2_–PCl_3_]^+^[PCl_6_]^−^, the cyclization leads to the targeted quantity of HCP with a small amount of cyclic tetramer and pentamer impurities (Figure 6a–d). Due to the fact that, not only does the target interaction of the linear cation [Cl(PCl_2_=N)_2_–PCl_3_]^+^ with HMDS (or NH_4_Cl) lead to the formation of HCP, but also the side reaction of HMDS with the PCl_5_ molecule from the [PCl_6_]^−^ counterion leads first to the formation of a Cl_3_P=NSiMe_3_ monomer and its further spontaneous growth with immediate cyclization in the presence of HMDS excess, the total yield of HCP turns out to be more than the theoretical 100%. For the higher cycles, the experimental data of cyclization product composition always show some deviation from the theoretical calculation, so the broadening of molecular weight distribution occurs (Table 4). That is most probably caused by side reactions, including partial intermolecular cross-linking (Figure 3b) and HMDS interaction with PCl_5_ from counterion, leading to spontaneous formation of phosphoraninime Cl_3_P=NSiMe_3_ (Figure 3c) with the following polymerization and increase in chain length (Figure 3d). For example, in the case of the expected final product with five -PCl_2_=N- units, one can see the predominant content of tetramer and high content of cyclophosphazenes with 9 and 10 units (Figure 6e–h).

This is due to the fact that some of the reagent molecules with four units took part in the formation of longer chains, which resulted in a decrease in pentamer yield. Similar behavior is observed when using [Cl(PCl_2_=N)_7_PCl_3_]^+^[PCl]^6−^ for the preparation of cyclooctaphosphazene (Figure 7).

In summary, the linear phosphazenes with number of chain units higher than five are more disposed to intermolecular reactions than short linear oligomers. With the increase in chain length, the cyclization becomes more hindered and the yield of cyclic product decreases. In case of cyclic product with calculated chain length of eight units, the broadened signal in range of −19 ppm, characteristic for backbone units of linear phosphazenes with polymerization degree about 10–20, is observed.

Table 4 indicates the increasing intermolecular reactions between oligomers [Cl(PCl_2_=N)_n_–PCl_3_]^+^[PCl]^6−^ (where *n* ≥ 5) in comparison with cyclization reaction. According to MALDI-TOF data of phenoxy derivatized compounds, the product contains cyclic oligomers with the number of chain units up to 12. On the other hand, the linear phosphazenes were not observed on spectra, which is an indicator of their high molecular weight.

## 6. Conclusions

A new method for the synthesis of cyclic phosphazenes via the interaction of corresponding linear chains with HMDS or NH_4_Cl was developed. HCP can be obtained in a yield higher than 100% of the theoretical value. At the same time, higher cycles with 4–7 units can be prepared as dominant products, with neighbor homologues as the main side-products.

## Figures and Tables

**Figure 1 ijms-22-05958-f001:**
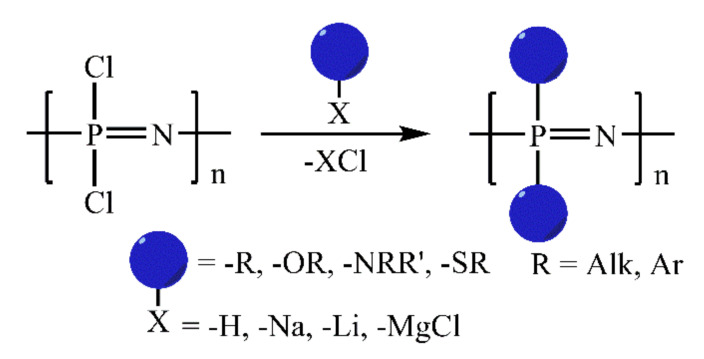
Nucleophilic substitution of chlorine atoms in dichlorophosphazenes.

**Figure 2 ijms-22-05958-f002:**
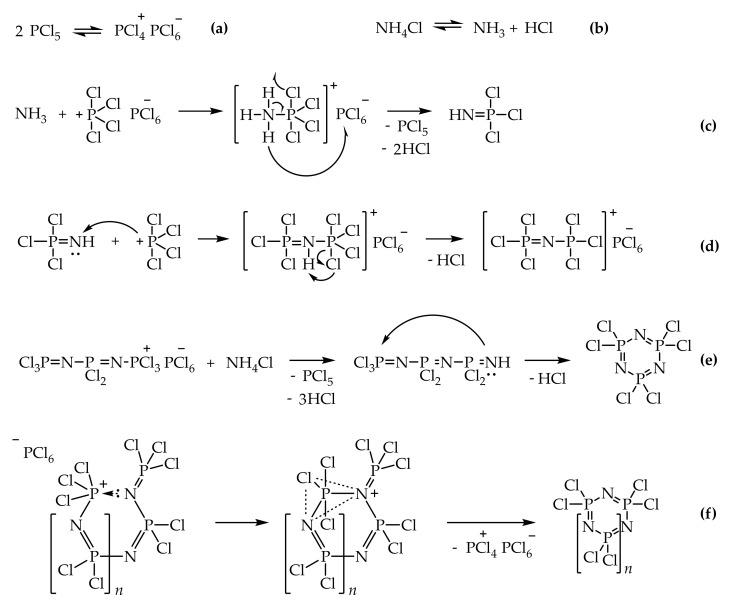
The mechanism of the formation of cyclic chlorophosphazenes proposed by Becke-Goehring et al. [36] (**a**–**e**), and alternative cyclization mechanism proposed by Emsley et al. (**f**) [37].

**Figure 3 ijms-22-05958-f003:**
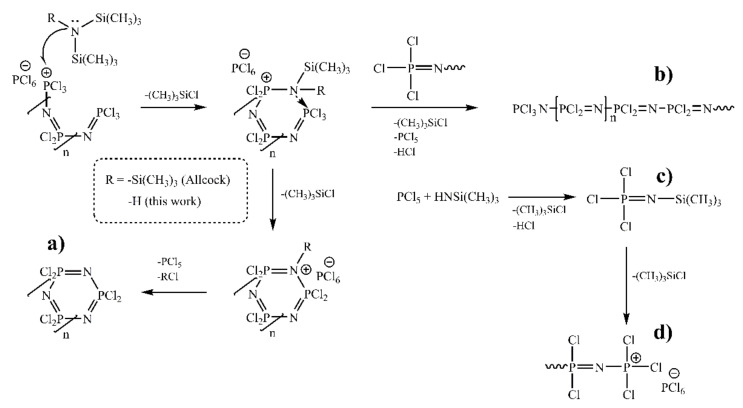
The formation of various-sized cyclophosphazenes via short-chain linear oligodichlorophosphazenes [Cl(PCl_2_=N)_n_–PCl_3_]^+^[PCl_6_]^−^ cyclization (**a**) and possible side reactions (**b**–**d**).

**Figure 4 ijms-22-05958-f004:**
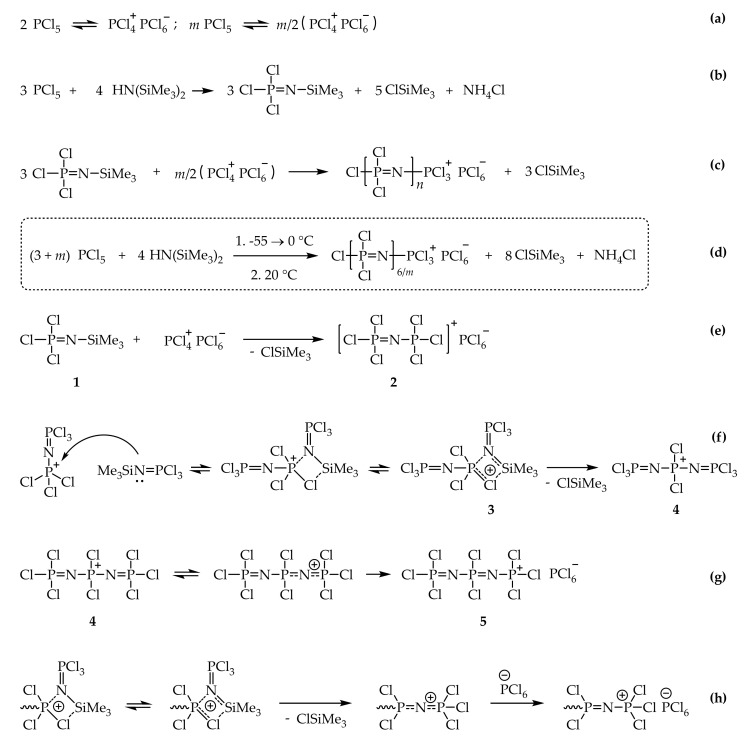
One-pot synthesis of oligodichlorophosphazene from PCl_5_ and HMDS (**d**) via living cationic process and its mechanism: initiator formation (**a**), monomer formation (**b**), general scheme for polymerization (**c**), formation of active centers and initiation (**e**–**g**), chain growth (**h**).

**Figure 5 ijms-22-05958-f005:**
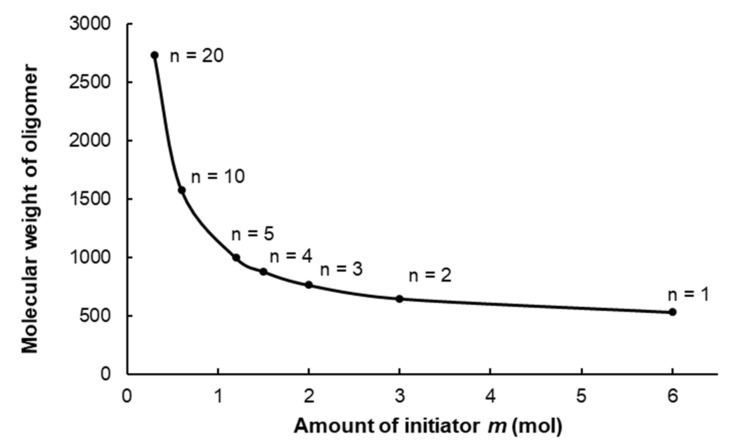
Dependence of molecular weight of linear oligodichlorophosphazene [Cl(PCl_2_=N)_n_PCl_3_]^+^[PCl_6_]^−^ from amount of initiator PCl_5_.

**Figure 6 ijms-22-05958-f006:**
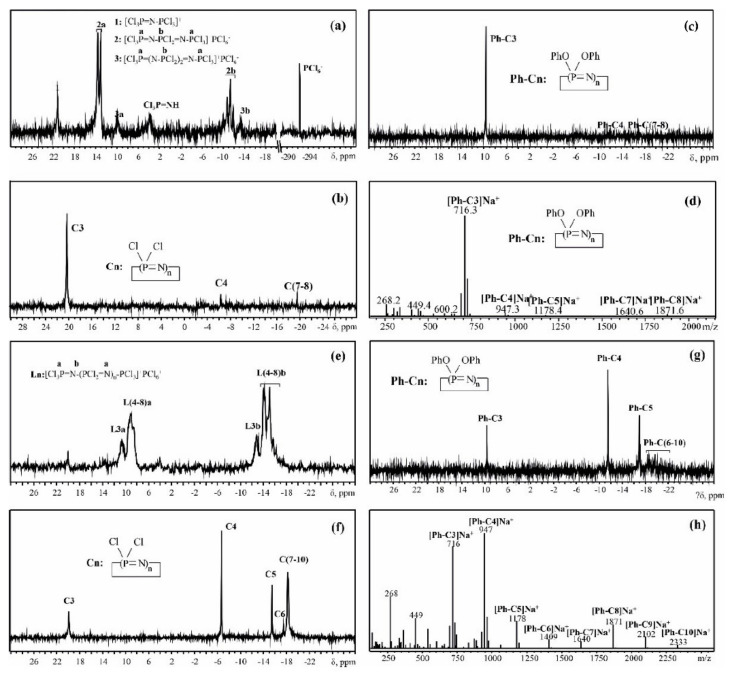
^31^P NMR spectra of linear oligophosphazene: [Cl(PCl_2_=N)_2_PCl_3_]^+^[PCl_6_]^−^ (**a**) and [Cl(PCl_2_=N)_4_PCl_3_]^+^[PCl]^6−^ (**e**); product of their cyclization (**b**,**f** respectively), phenoxylated derivatives of cyclization products (**c**,**g** respectively) and MALDI-TOF spectra of phenoxylated products (**d**,**h** respectively).

**Figure 7 ijms-22-05958-f007:**
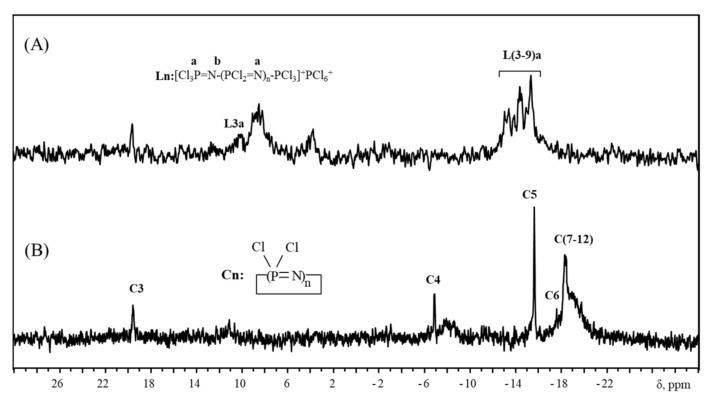
^31^P NMR spectra of linear oligophosphazene [Cl(PCl_2_=N)_7_PCl_3_]^+^[PCl_6_]^−^ (**A**) and its cyclization product (**B**).

**Table 1 ijms-22-05958-t001:** ^31^P NMR chemical shifts of chlorocyclophazenes.

Value of ^31^P NMR Chemical Shift of Chlorophosphazenes [Cl_2_P=N]*_n_*				Reference
5	6	7	≥8	
−17.0	−16.0	−18.0	-	[35] (1972)
−15.1	−15.3	−17.0	–17.7	[39] (2014)

**Table 2 ijms-22-05958-t002:** Amounts of HMDS for synthesis of linear oligodichlorophosphazenes with general formula [Cl(PCl_2_=N)_n_PCl_3_]^+^[PCl_6_]^−^.

n	Weight (g)	mol	Yield of Product (%)
2	2.58	0.016	75
6	3.88	0.024	82
7	4.03	0.025	84
9	4.23	0.027	88

**Table 3 ijms-22-05958-t003:** Amounts of HMDS for cyclization of linear oligodichlorophosphazenes with general formula [Cl(PCl_2_=N)_n_PCl_3_]^+^[PCl_6_]^−^.

n	Weight (g)	mol	Yield of Product (%)
2	1.49	0.00923	130 ^1^
6	0.87	0.00539	80
7	0.79	0.00488	78
9	0.66	0.00410	71

^1^ The formation of hexachlorocyclotriphosphazene is resulted not only after reaction of HDMS with [Cl(PCl_2_=N)_n_PCl_3_]^+^[PCl_6_]^−^ but also after its reaction with [PCl_6_]^−^ ion.

**Table 4 ijms-22-05958-t004:** The composition of cyclic phopshazenes obtained by cyclization of linear oligomers.

Cycle Size *k* of the Cyclic Homologue in the Product	*m/z* of Phenoxylated Derivatives [N=P(OPh)_2_]*_k_*	PCl_5_: HMDS Ratio Used to Obtain Linear Chlorophosphazene Oligomer/Calculated Average Number of P=N-Links in the Product after Cyclization		
		2:3/*k* = 3	8:9/*k* = 5	1.04:1/*k* = 8
		Composition of the Cyclic Product, Determined by MALDI-TOF ^a)^		
3	694	95.00	38.40	0.7
4	925	3.00	40.30	0.8
5	1156	-	7.81	3.4
6	1387	0.50	2.32	47.9
7	1618	0.50	1.69	31.7
8	1849	1.00	5.70^)^	7.4^)^
9	2080	-	2.95	3.1
10	2311	-	0.84	2.7
11	2541	-	-	1.7
12	2773	-	-	0.5

^a)^ Determined from absolute intensities of the oligomer species on the MALDI-TOF spectra of phenoxy-derivatized cyclophosphazenes [N=P(OPh)_2_]_k_.

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
