# Peer review of "Novel Approach for the Synthesis of Chlorophosphazene Cycles with a Defined Size via Controlled Cyclization of Linear Oligodichlorophosphazenes [Cl(PCl2=N)n–PCl3]+[PCl6]−"

_ijms, 2021, doi:10.3390/ijms22115958_

Round 1
Reviewer 1 Report
In this paper the authors describe a new variant of the synthesis of cyclic oligodichlorophosphazenes which allows to control their average polymerization degree. They also postulate the mechanism of the cyclization of linear oligophosphazenes in the light of their results. The research is original and its results are valuable. Although this paper is worthy to be published in Journal of Molecular Sciences I do not see any reason for its printing in the form of communication. It describes an extensive research, it contains spacious introduction being an overview on mechanisms of the polydichlorophosphazenes formation and includes about 50 cited articles. Important results are hidden in supporting information. This paper should be published as regular article containing experimental part and some more results transferred from supporting information.
Other remarks:
There is a big disorder in citations. For example in line 47 is citation [31], which is followed by [44] in line, 51. Citations [32]-[43] are after [46]. All citation in the text should be put in appropriate order and all should be shown in the text.
line 44. flammability should be changed to inflammability or non-flammability.
Figure 3 equation b, coefficient 4 should be placed before HMDS.
Figure 3. Equation f, structure 3, Arrow should be reversed. This is nucleophilic N which attack electrophilic P.
Line 183. „the formation of intermediate 3“. Intermediate appears in a minimum on the reaction free enthalpy diagram. Structure 3 formation requires a large negative entropy input. The formation of intermediate is doubtful, although it may be an intermediate structure.
line 255. should be „phenoxylated product (D)“
Author Response
|
This paper should be published as regular article containing experimental part and some more results transferred from supporting information.
|
We thank the distinguished reviewer for valuable comments and additions. The manuscript has been revised. The supporting information file was removed because all its content was moved to the manuscript body. |
|
There is a big disorder in citations. For example in line 47 is citation [31], which is followed by [44] in line, 51. Citations [32]-[43] are after [46]. All citation in the text should be put in appropriate order and all should be shown in the text. |
Fixed. The references and bibliography were revised using reference manager. |
|
Line 44. Flammability should be changed to inflammability or non-flammability. |
Fixed |
|
Figure 3 equation b, coefficient 4 should be placed before HMDS |
Fixed |
|
Figure 3. Equation f, structure 3, Arrow should be reversed. This is nucleophilic N which attack electrophilic P. |
Fixed |
|
Line 183. „the formation of intermediate 3“. Intermediate appears in a minimum on the reaction free enthalpy diagram. Structure 3 formation requires a large negative entropy input. The formation of intermediate is doubtful, although it may be an intermediate structure. |
The sentence was slightly revised. |
|
Line 255. Should be „phenoxylated product (D)“ |
Fixed |

Reviewer 2 Report
I would encourage the authors to include GPC data of the oligomer. Also provide some comparison of physical properties of linear and cyclic molecules, such as viscosity, Tg, etc.
Author Response
|
I would encourage the authors to include GPC data of the oligomer. Also provide some comparison of physical properties of linear and cyclic molecules, such as viscosity, Tg, etc. |
We thank the distinguished reviewer for valuable comments. We believe that this information is not required in this article, since all described oligomers have already been previously extensively studied and described by other researchers, including, for example: 1. Stokes On Trimetaphosphimic Acid and Its Decomposition-Products.; Amer. Chem. J, 1896; 2. Gleria, M.; Jaeger, R.D. Phosphazenes: A Worldwide Insight; Nova Publishers, 2004; ISBN 978-1-59033-423-2. 3. Becke‐Goehring, M.; Lehr, W. Über Phosphor-Stickstoff-Verbindungen. XVI. Die Synthese der Phosphornitrid-dichloride. Zeitschrift für anorganische und allgemeine Chemie 1964, 327, 128–138, doi:https://doi.org/10.1002/zaac.19643270305. 4. Emsley, J.; Udy, P.B. Elucidation of the Reaction of Phosphorus Pentachloride and Ammonium Chloride by Phosphorus-31 Nuclear Magnetic Resonance Spectroscopy. J. Chem. Soc. A 1970, 3025–3029, doi:10.1039/J19700003025. 5. Lund, L.G.; Paddock, N.L.; Proctor, J.E.; Searle, H.T. 514. Phosphonitrilic Derivatives. Part I. The Preparation of Cyclic and Linear Phosphonitrilic Chlorides. J. Chem. Soc. 1960, 2542–2547, doi:10.1039/JR9600002542. 6. H.R. Allcock. A New Route to the Phosphazene Polymerization Precursors, Cl3PNSiMe3 and (NPCl2)3 | Inorganic Chemistry Available online: https://pubs.acs.org/doi/abs/10.1021/ic980534p (accessed on 11 May 2021). 7. D. J. Bowers, B. D. Wright, V. Scionti, A. Schultz, M. J. Panzner, E. B. Twum, L.-L. Li, B. C. Katzenmeyer, B. S. Thome, P. L. Rinaldi, C. Wesdemiotis, W. J. Youngs, C. A. Tessier, Inorganic Chemistry 2014, 53, 8874-8886. 8. I. S. Sirotin, Y. V. Bilichenko, O. V. Suraeva, A. N. Solodukhin, V. V. Kireev, Polymer Science Series B 2013, 55, 63-68; bL. Wang, Y. Ye, Z. Ju, S. Zhong, Y. Zhao, Phosphorus, Sulfur, and Silicon and the Related Elements 2009, 184, 1958-1963. We just suggested an alternative way to synthesize them. In our work, we used a combination of NMR and MALDI-TOF, which in general, in our opinion, is a sufficient characterization for these types of compounds since results obtained are fully consistent with above mentioned studies. |

Round 2
Reviewer 1 Report
Authors satisfactorily considered my remarks